# Tumor Mutational Burden in Cervical Cancer as Potential Marker for Immunotherapy Responders

**DOI:** 10.3390/cancers17182963

**Published:** 2025-09-10

**Authors:** Magdalena Ewa Kowalkowska, Katarzyna Kamińska, Joanna Wojtysiak, Krzysztof Koper, Adrianna Makarewicz, Bronisława Pietrzak, Dorota Bomba-Opoń, Marzena Dębska, Mirosław Wielgoś, Marek Grabiec, Marzena Anna Lewandowska

**Affiliations:** 1Clinical Department of Oncological Gynecology, Oncology Centre in Bydgoszcz, 85-796 Bydgoszcz, Poland; 2Thoracic Research Centre, Collegium Medicum, Nicolaus Copernicus University, 85-067 Bydgoszcz, Poland; 3Department of Genetics and Molecular Oncology, Oncology Centre in Bydgoszcz, 85-796 Bydgoszcz, Poland; kaminskak@co.bydgoszcz.pl (K.K.); wojtysiakj@co.bydgoszcz.pl (J.W.); 4Clinical Department of Oncology, Oncology Centre in Bydgoszcz, 85-796 Bydgoszcz, Poland; koperk@co.bydgoszcz.pl (K.K.); makarewicza@co.bydgoszcz.pl (A.M.); 5Department of Oncological Surgery, Nicolaus Copernicus University in Toruń, Collegium Medicum, 85-067 Bydgoszcz, Poland; 6Department of Obstetrics and Perinatology, National Medical Institute of the Ministry of Interior and Administration, 02-507 Warsaw, Poland; bronislawa.pietrzak@pimmswia.gov.pl (B.P.); dorota.bomba@pimmswia.gov.pl (D.B.-O.); marzena.debska@pimmswia.gov.pl (M.D.); miroslaw.wielgos@pimmswia.gov.pl (M.W.); 7Department of Clinical Medicine, Bydgoszcz University of Science and Technology, 85-796 Bydgoszcz, Poland; grabiecm@cm.umk.pl; 8Clinical Department of Thoracic Surgery and Tumors, Collegium Medicum UMK, 85-796 Bydgoszcz, Poland

**Keywords:** cervical cancer, tumor mutational burden, next-generation sequencing, HPV, targeted therapy, genomic profiling, somatic mutations, immunotherapy biomarkers, FIGO staging, gynecologic oncology

## Abstract

Cervical cancer remains a significant health burden worldwide, and identifying biomarkers that predict response to immunotherapy is a growing research priority. Tumor mutational burden (TMB), which reflects the number of somatic mutations in a tumor, has been proposed as one such marker. In this study, we analyzed tumor samples from 61 patients with cervical cancer using next-generation sequencing. High TMB was found in over half of the cases and was associated with specific clinical features such as nodal involvement, diabetes, and infection with HPV52. However, TMB did not correlate with disease stage or the most common HPV types. Our results suggest that TMB may help refine patient selection for immunotherapy in cervical cancer but further research is warranted.

## 1. Introduction

Cervical cancer remains a major global health issue, primarily driven by persistent human papillomavirus (HPV) infection [1,2,3,4]. Although immunotherapy, particularly immune checkpoint inhibitors targeting the PD-1/PD-L1 pathway, has shown promise in treating recurrent or metastatic cervical cancer [5,6,7], many patients do not respond to these treatments [8,9]. Therefore, predictive biomarkers are crucial for identifying patients who may benefit from immunotherapy.

Tumor mutational burden (TMB), which quantifies the total number of somatic mutations per megabase of tumor DNA, has emerged as a potential biomarker in various cancers [10,11]. Higher TMB is thought to correlate with increased neoantigen production, enhancing immune recognition and response to immunotherapies [12,13]. While cervical cancer generally exhibits a lower TMB compared to cancers like melanoma or non-small cell lung cancer, subsets of patients with higher TMB may respond better to immune checkpoint inhibitors [14,15].

The role of TMB as a biomarker in cervical cancer is still under investigation [16,17]. Its interpretation is particularly complex due to the interplay of HPV-induced mutagenesis, other etiological factors like smoking, and a unique inflammatory tumor microenvironment, all of which can influence the mutational landscape. Current challenges include establishing standardized TMB thresholds and accounting for variability in measurement techniques. Additionally, the tumor microenvironment and other factors such as PD-L1 expression influence immunotherapy outcomes [18]. Future research must focus on refining TMB’s role in predicting immunotherapy response, possibly by integrating it with other immune-related biomarkers to improve treatment stratification and outcomes. In the meanwhile, the current analysis attempts to profile patients’ baseline and genetic characteristics depending on the presence of TMB [15].

## 2. Materials and Methods

### 2.1. Study Population

This retrospective study included 61 consecutive female patients (mean age 55.1 ± 14.0 years) diagnosed and treated at the Clinical Department of Oncological Gynecology at the Bydgoszcz Oncology Center between 2014 and 2021 who consented to and underwent tumor tissue genomic profiling by next-generation sequencing (NGS). Informed consent was obtained from each participant, and the study was approved by the Institutional Bioethics Committee (approval no. [KB265/2012]).

### 2.2. Sample Collection and DNA Isolation

Tumor specimens were obtained through various clinical procedures including diagnostic biopsies, surgical resections (hysterectomy or trachelectomy), conization, or endocervical curettage, depending on clinical indications and disease stage. Only formalin-fixed, paraffin-embedded (FFPE) tumor blocks with a tumor cell content between 5% and 90% were accepted for downstream analysis. While quality control filters were used, we did not perform purity-based normalization.

DNA was extracted from 5–10 μm tissue sections using standard protocols optimized for FFPE material. The extracted DNA was quantified using the Quantifluor^®^ dsDNA System (Promega, Madison, WI, USA) and further evaluated for integrity and fragmentation using the DNA Fragmentation Quantification Assay (Entrogen, Thousand Oaks, CA, USA).

### 2.3. NGS Library Preparation and Sequencing

Two targeted NGS panels were used for library construction: ONCOaccuPanel™ (NGeneBio, Seoul, Republic of Korea), a comprehensive 344-gene panel covering key oncogenic pathways including cell cycle regulation, DNA damage repair, and PI3K signaling, and BRCAaccuTest PLUS™ (NGeneBio, Seoul, Republic of Korea), a BRCA1/2-focused panel included to assess HRD-related genes, as part of exploratory analysis. Library preparation was performed according to the manufacturer’s instructions using hybrid capture enrichment. Paired-end sequencing (2 × 150 bp) was conducted on either the MiniSeq or NextSeq 550 platform (Illumina, San Diego, CA, USA), depending on batch size and sample availability.

### 2.4. Bioinformatic Processing and Variant Annotation

Primary sequence data (FASTQ) were processed using NGeneAnalySys^®^ software version 1.8.0.3 (NGeneBio), which includes alignment to the human reference genome (GRCh37), variant calling, and annotation of single-nucleotide variants (SNVs), insertions/deletions (indels), and copy number alterations. Variant interpretation followed ACMG/AMP 2015 criteria [19] for pathogenicity classification. Tumor mutational burden (TMB) was automatically calculated by NGeneAnalySys^®^ as the total number of non-synonymous somatic mutations per megabase of targeted exonic sequence. The threshold (23 Mut/Mb) was internally validated for NGS and was different from the FDA’s commonly used ≥10 Mut/Mb based on whole-exome sequencing. The difference arises from variations in panel size (which covers 0.64 Mb of exonic regions as a 344-gene panel, inherently detecting a large number of variants, including numerous VUS) and target region content. The NGS threshold was used for sensitivity analysis. The TMB calculation in ONCOaccuPanel™ targeted 1.29 Mb of exonic regions. Samples were stratified into TMB-high (≥23 mutations/Mb) and TMB-low groups, consistent with current FDA thresholds for immune checkpoint inhibitor eligibility [20,21].

Microsatellite instability (MSI) in NGS is assessed using multiple conditions: (i) a total number of SNV/INDEL mutations across all genes of >30, an INDEL Index of >20%, and additional considerations, including the presence of AMC MSI marker, detection of pathogenic MMR gene mutations, evaluation of tumor purity, filtering of germline variants, false positives/negatives, and POLE pathogenic variants.

### 2.5. Statistical and Comparative Genomic Analyses

The comparative mutational analysis between ca. planoepitheliale and adenocarcinoma subtypes was visualized using oncoplots generated via custom scripts in R (maftools v2.10.0). To evaluate co-mutation patterns and mutual exclusivity, a somatic interactions plot was used to depict the conditional probability of observing pathogenic variants in one gene given mutations in another. Genes with pathogenic or likely pathogenic variants (as per ACMG guidelines) were included in this matrix. To investigate potential clinical correlates of TMB status, patients were stratified into TMB-high and TMB-low groups. Continuous variables were compared using the Mann–Whitney U test, and categorical variables were assessed via the chi-squared or Fisher’s exact test. A univariable logistic regression analysis was performed to identify predictors of high TMB status, including histological type, tumor grade, and mutational signatures. All statistical tests were two-tailed and considered significant at *p* < 0.05. Analyses were performed using R software (v4.2.2) and GraphPad Prism (v9.0).

## 3. Results

### 3.1. Baseline Characteristics and Treatment Modalities

Table 1 lists the patients’ baseline characteristics, divided into TMB+ and TMB− cohorts: Among the 61 patients included in this study (mean age: 55.1 ± 14.0 years), 50 (82.0%) were diagnosed with ca. planoepitheliale and 11 (18.0%) with adenocarcinoma. According to FIGO staging, the majority of cases were classified as stage IB1 (32%), followed by IB2, IIA1, and IIIA (see Figure 1A,B).

Surgical treatment was performed in 52 patients (85.2%), including radical hysterectomy or trachelectomy. Postoperative complications included two in-hospital deaths (3.3%), one case requiring nephrostomy, and three temporary or permanent colostomies. Adjuvant therapies were commonly administered: 52 patients (85.2%) underwent brachytherapy, and 50 (82.7%) received external beam radiotherapy (EBRT), often in combination with surgery. Chemotherapy was administered in 27 cases (44.3%), 20 of which (74.0%) were administered concurrently with surgery (Figure 1C,D).

### 3.2. Genomic Landscape and Somatic Mutations

NGS analysis identified a diverse spectrum of somatic alterations. There was no marked difference in the mutation classes between ca. planoepitheliale and adenocarcinoma in the analyzed subset: missense mutations accounted for 74% vs. 79%, frame shift deletions for 13% and 10%, nonsense mutations for 5.5% and 3.7%, and frame shift insertions for 4% and 2.9%, respectively (Figure 2A). Single-nucleotide variants (SNVs) were the most predominant mutation types, accounting for 83% and 80% of the mutations within ca. planoepitheliale and adenocarcinoma, respectively, and were followed by deletions (12% vs. 14%), insertions (3.1% vs. 4%), and other complex types (1.8% vs. 1.8%), without marked differences (Figure 2B).

Single-base substitution (SBS) analysis revealed a predominance of A>G and T>C transitions (A>G 30% vs. 37% and T>C 31% vs. 32% for ca. planoepitheliale and adenocarcinoma, respectively), which are typical of DNA aging and APOBEC enzyme activity, particularly in the context of viral infections such as HPV (Figure 2C) resembling in part COSMIC Signatures 2 and 13 (C>A, C>G, C>T, T>A, T>C, T>G comprising 3.5%, 3.5%, 5.2%, 2.2%, 31%, 1.5% and 1.8%, 1.8%, 2.3%, 3.2%, 32%, 2.3% for ca. planoepitheliale and adenocarcinoma). However, again, there were no marked differences between them. Variants classified as pathogenic or likely pathogenic (≥1 ACMG evidence star) were detected across TIER 1 and TIER 2 genes. The most frequently altered genes included PIK3CA (60%), KRAS (13%), PTEN (6.7%), NRAS (6.7%), IDH1 (6.7%), and BRAF (6.7%) in ca. planoepitheliale; KRAS mutations (42%) followed by PIK3CA, TP53, MAP2K1, GNAS, ERBB2, PTEN, and NRAS (all 8.3%) were noted in adenocarcinoma (Figure 2D).

Somatic interaction analysis (Figure 3) revealed that PIK3CA and ARID1A mutations frequently co-occurred, especially in high-TMB samples, suggesting shared oncogenic pathways. The co-occurrence matrix also highlighted potentially mutually exclusive relationships, such as between TP53 and PTEN, though statistical significance was not reached (*p* > 0.1, FDR-adjusted).

Figure 3 shows the mutational distribution and burden stratified by histological subtype. Notable proportions of PIK3CA:p.E453Q and ARID1A:p.Q487SfsTer132 mutations, previously described in colorectal or endometrial carcinomas, were detected in ca. planoepitheliale tumors here but are not canonically linked to cervical cancer pathogenesis (CIViC, OncoKB). Similarly, TP53:p.E171del, a variant affecting the DNA binding domain, was recurrently observed but is not universally considered pathogenic (CIViC 2771). Multiple PTEN and KRAS variants were downgraded to TIER 2 or were without clear evidence of pathogenicity.

### 3.3. Tumor Mutational Burden and Microsatellite Instability

High TMB, defined as ≥10 mutations/Mb, was detected in 36 patients (59.0%), while microsatellite instability (MSI-high) was found in only 2 patients (3.3%). Our observed MSI rate (3.3%) is consistent with TCGA data (<5%) and reflects HPV-driven rather than mismatch repair-driven tumorigenesis. The TMB distribution was not significantly different between the ca. planoepitheliale and adenocarcinoma types (high TMB: 12% vs. 22%, respectively; *p* = 0.308), nor was it associated with advanced FIGO stage (*p* = 0.519) or HPV 16/18 status (20% vs. 26%; *p* = 0.309) (Figure 1C,D). Despite retrospective analysis, no patients in this cohort were eligible for immunotherapy at the time of treatment.

Patients with nodal involvement had a significantly higher mean TMB (+13 mutations/Mb). TMB was also elevated in patients with HPV52 infection (+25 mutations/Mb), Type 2 diabetes mellitus (+10), HPV68 positivity (+5.8), or grade G3 (+3.6 vs. G2). In contrast, hypertension was associated with a decreased TMB (−5.5 mutations/Mb). The results of the univariable analysis of TMB and clinical features are available in Figure 4.

Clinically, patients are now followed for a median of 6 years [IQR; 5–7], with an overall survival (OS) rate of 95.2% and progression-free survival (PFS) rate of 90.5%, without differences between high and low TMB (Appendix A Figure A1).

### 3.4. Homologous Recombination Deficiency

HRD, defined as LOH ≥ 16, was identified in 34 cases (55.7%) overall, with 25 occurring in the TMB-high group and 9 in TMB-low (Figure 5B). No statistically significant association was observed between HRD status and TMB category (*p* = 0.291).

The top five most frequent somatic variants included recurrent non-coding and coding alterations (Figure 5A). Only the variant chr22:g.29695793:A>C, proximal to CHEK2, showed a numerical trend toward HRD enrichment. Other recurrent loci such as NG_012113.2:g.74781G>A (PIK3CA) and NG_011536.1:g.51312G>T (ARID1A) were observed across both TMB strata without clear distributional bias. Figure 5C,D demonstrate sensitivity analysis for TMB threshold of 23 Mut/Mb as internally validated for NGS.

## 4. Discussion

The management of advanced cervical cancer (FIGO IIB and higher) remains particularly challenging due to the limited systemic options and frequent resistance to conventional therapies. Immune checkpoint inhibitors (ICIs) have transformed oncology by enhancing T-cell-mediated anti-tumor responses, and their application in cervical cancer has led to regulatory approval of pembrolizumab for PD-L1-positive advanced cases, based on results from KEYNOTE-158 [5,22].

### 4.1. Tumor Mutational Burden and Its Clinical Relevance

Tumor mutational burden (TMB), reflecting the number of somatic mutations per megabase (mut/Mb), is a genomic biomarker associated with neoantigen load and potential immunogenicity [10]. TMB can be assessed using a number of NGS platforms, including whole-genome sequencing (WGS), WES, or targeted panel sequencing. WES is the “gold standard” for measuring TMB [23], allowing for the detection of somatic coding (non-synonymous) mutations present within the entire exome. WES targets ~30 Mb of coding regions, covering all ~22,000 genes and making up ~1% of the genome [15]. The definition of TMB varies with the measurement method utilized [24]. While TMB has been accurately measured by WES in several studies, this is currently not feasible in clinical practice due to its high cost, relatively long turnaround time, and need for sufficient tissue samples [10]. Multiple commercially available gene panels designed for TMB estimation cover between approximately 0.80 and 2.40 Mb, representing <5% of the total coding sequence (35–44) [25]. Their use has been validated across several malignancies, including NSCLC and melanoma, where high TMB correlates with increased response to ICIs [11,26]. In cervical cancer, the FDA has approved high TMB (≥10 mut/Mb) as a pan-cancer biomarker for immunotherapy, including for advanced disease [21]. Panel-based TMB is becoming widely accepted, though harmonization across panels is ongoing [23]. Although direct correlation data for our specific panel in cervical cancer are limited, studies have demonstrated that large hybrid capture panels like the one used here show strong correlations with WES-derived TMB across various cancer types, providing a reliable and practical alternative for clinical assessment [23,25]. This alignment supports the validity of our TMB measurements. Our study confirmed that TMB-high status is present in a substantial subset (59.0%) of cervical cancer patients and shows associations with several clinical variables, including nodal involvement, HPV subtypes (notably HPV52), metabolic comorbidities (e.g., diabetes), and tumor grading. While limited to univariable analysis, similar observations have been made in other cancers, where high TMB has been linked to more aggressive clinical behavior, likely due to genomic instability [27,28].

However, it remains unclear whether high TMB alone predicts responsiveness to immunotherapy in cervical cancer. While a high TMB could theoretically enhance neoantigen presentation, not all patients with TMB-high tumors respond favorably to checkpoint inhibitors [29]. This discrepancy has prompted investigations into additional markers such as PD-L1 expression, tumor-infiltrating lymphocytes (TILs), microsatellite instability (MSI), and homologous recombination deficiency (HRD).

### 4.2. TMB and HRD: Disconnected Genomic Features

We investigated the relationship between TMB and HRD in our cohort, defining HRD positivity by a loss of heterozygosity (LOH) score ≥ 16. HRD was present in 14.8% of patients, with no significant difference in its distribution between the TMB-high and TMB-low groups (*p* = 0.291). This finding is consistent with prior studies suggesting that while both HRD and TMB reflect forms of genomic instability, they may arise from distinct biological mechanisms and are not necessarily correlated [15,27,29].

While HRD and pathogenic BRCA1/2 mutations are established predictive biomarkers for PARP inhibitor sensitivity in cancers such as ovarian and breast, their role in cervical cancer remains under investigation [6,30,31]. In our cohort, only a few canonical pathogenic BRCA1/2 mutations were identified. Although a high proportion of cases exhibited genomic scars suggestive of HRD, this finding currently lacks definitive causality and clinical utility for treatment selection in cervical cancer. Further clinical trials are needed to determine whether HRD status can predict the response to PARP inhibitors or other DNA-damaging agents in this patient population.

Among the most common somatic loci identified in our analysis (top fifth percentile), only one non-coding variant—chr22:g.29695793:A>C, near CHEK2—was enriched in HRD-positive cases. Other mutations of interest, such as PIK3CA (E545K, NG_012113.2:g.74781G>A) and ARID1A (R1324L, NG_011536.1:g.51312G>T), typically observed in noncervical cancers such as colorectal or endometrial tumors, were found; these variants may reflect shared pathways (e.g., APOBEC-driven mutagenesis) across tumor types. In addition, these were observed across TMB strata without a strong association with HRD, suggesting potential independence of these pathways in cervical tumorigenesis.

### 4.3. Clinical and Molecular Heterogeneity

The clinical phenotype of TMB-high tumors in our cohort included increased prevalence of nodal metastasis and association with rare HPV types, particularly HPV52 and HPV68. This is in line with previous studies in the literature reporting diverse oncogenic potential among HPV genotypes, which may influence genomic instability and immune recognition [32].

In contrast, patients with metabolic comorbidities such as hypertension exhibited lower TMB values. This inverse association has not been widely reported and arises from univariable analysis only, but systemic inflammation and immunosenescence in comorbid states may modulate tumor evolution and mutagenesis [33,34]. Other associations remain to be confirmed in multivariable models with larger sample sizes.

The most frequent genetic alterations identified herein included PIK3CA, KRAS, PTEN, and TP53 mutations, which are recurrently observed in cervical tumors and other gynecologic cancers [35]. These mutations not only are oncogenic drivers but also represent potential therapeutic targets.

### 4.4. Future Directions and Unanswered Questions

Despite promising results, our findings highlight the complexity of integrating TMB into clinical decision-making. While TMB is an FDA-approved biomarker for ICI therapy in advanced cervical cancer, its potential role as a selection tool for adjuvant or neoadjuvant immunotherapy following surgery remains speculative. Further randomized trials are needed to assess whether adjuvant immunotherapy can improve outcomes in TMB-high patients after surgical resection. For the time being, MSI remains the only currently validated marker.

Moreover, our study underscores the importance of contextualizing TMB with other biomarkers. For example, the interplay among TMB, PD-L1, MSI, HRD, and TILs could form the basis for comprehensive immuno-genomic profiling to stratify patients for treatment [36,37]. We recommend caution when interpreting samples with <20% tumor content, especially for INDEL-heavy assessments like MSI. While quality control filters were used, we did not perform purity-based normalization.

Advances in NGS technology, including whole-exome and panel-based sequencing, allow for rapid and cost-effective TMB assessment. However, standardization of assays and interpretation frameworks is necessary to ensure reproducibility and cross-study comparison [15].

## 5. Conclusions

TMB represents a promising yet complex biomarker in cervical cancer. While we observed clinical and genomic patterns associated with high TMB, we did not identify significant correlations with HRD. As cervical cancer therapy continues to evolve toward precision medicine, integrated genomic profiling—including TMB, HRD, and immune biomarkers—will be essential for optimizing treatment strategies.

## Figures and Tables

**Figure 1 cancers-17-02963-f001:**
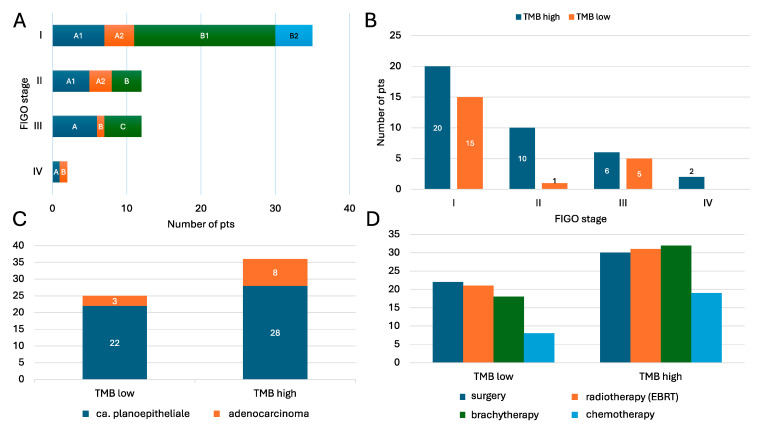
Baseline characteristics: FIGO stage (**A**,**B**), histological type (**C**), and treatment (**D**) according to TMB status (high vs. low). TMB, tumor mutational burden; EBRT, electron beam radiotherapy.

**Figure 2 cancers-17-02963-f002:**
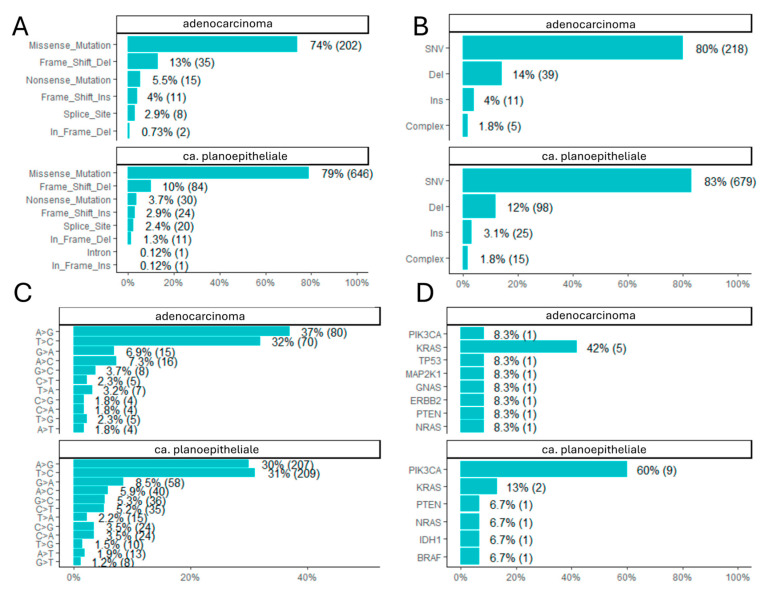
Somatic mutations analysis: (**A**): mutation classes (missense, frameshift, nonsense, etc.), (**B**): variant type distribution (SNV, indels, complex), (**C**): base substitution patterns, and (**D**): most frequently altered genes, comparing ca. planoepitheliale and adenocarcinoma cases.

**Figure 3 cancers-17-02963-f003:**
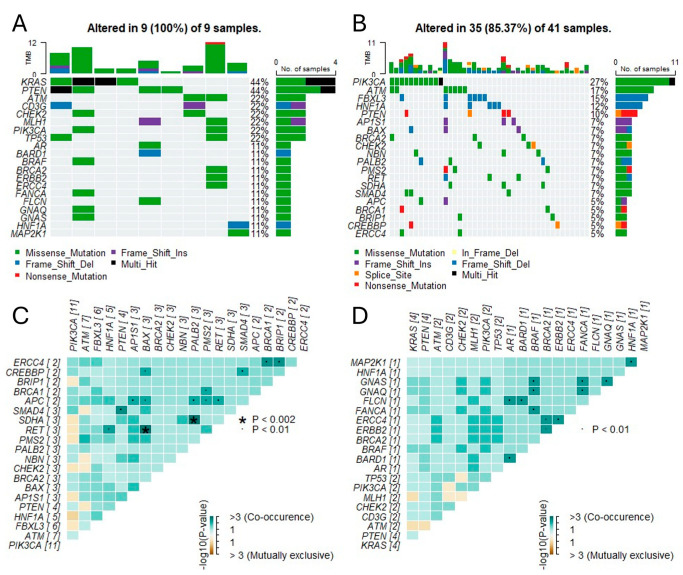
(**A**,**B**): Top 20 gene mutation frequencies identified through the NGS panel within ca. planoepitheliale (**A**) and adenocarcinoma (**B**). Each column represents a single sample, while the color of the rectangle indicates the type of mutation in a given gene for that sample. Only pathogenic variants according to ACMG are shown. The term Multi_Hit (black color) indicates that a sample had more than one type of mutation in a given gene. At the top of the plot, the bars represent the TMB calculated for each sample, with colors in each bar showing the proportional contribution of each mutation type to the total TMB of the sample. The bar plot on the right displays the percentage frequency of mutations in a given gene across all samples, with colors within each bar representing the proportional contribution of each mutation type. (**C**,**D**): Somatic interactions of PIK3CA and KRAS mutations. (**C**): In the top 20 most common genes within the ca. planoepitheliale cases, PIK3CA mutations did not co-occur with other mutations in a statistically significant manner and were mutually exclusive with mutations in HNF1A (*p* < 0.05). (**D**): In the top 20 most common genes within the adenocarcinoma cases, KRAS mutations did not co-occur with other mutations in a statistically significant manner, while they were mutually exclusive with mutations in MLH1 and ATM (*p* < 0.05) [1,2,3,4,5,6,7,11].

**Figure 4 cancers-17-02963-f004:**
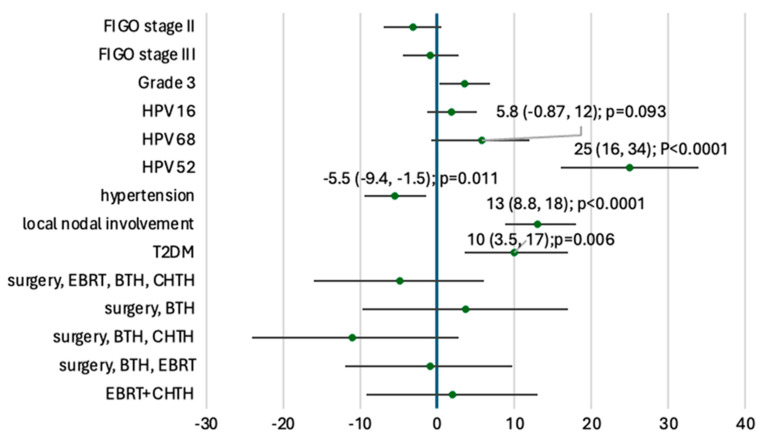
Univariable analysis: clinical predictors of high TMB. Leftward direction of the point estimate suggests a negative effect, e.g., low TMB; rightward deviated variable point estimates correlate with high TMB. Point estimates are mean changes with 95% confidence intervals. T2DM, type 2 diabetes mellitus; EBRT, electron beam radiotherapy; BTH, brachytherapy; CHTH, chemotherapy.

**Figure 5 cancers-17-02963-f005:**
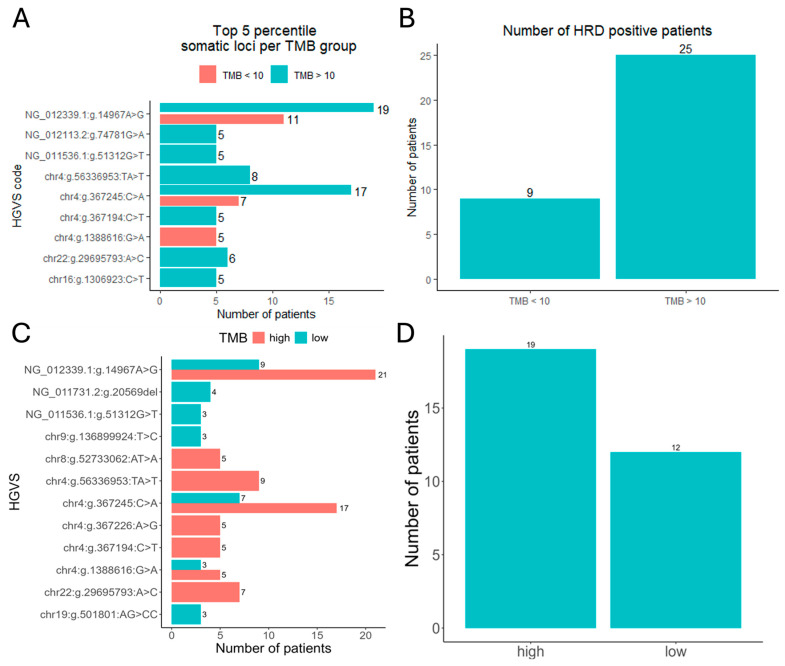
Top somatic mutations by TMB and HRD status. (**A**): Distribution of somatic mutations in top 5 most frequent across TMB-low vs. TMB-high groups (as defined by FDA criteria [10 Mut/MB]) with clinically significant mutations (e.g., NG_012113.2:g.74781G>A in PIK3CA) or non-coding/chromosomal variants (e.g., chr22:g.29695793.A>C near CHEK2). Mutation prevalence (number of patients) for selected variants in each TMB group. (**B**): Number of HRD-positive patients stratified by mutation frequency. (**C**,**D**): Sensitivity analysis for TMB threshold of 23 Mut/Mb as internally validated for NGS.

**Table 1 cancers-17-02963-t001:** Patients’ baseline characteristics.

Variable	All	TMB High	TMB Low
Mean age (±SD)	50.6 ± 14.0	51.1 ± 14.6	49.8 ± 13.4
Ca. planoepitheliale	50 (82.0%)	28 (77.8%)	22 (88.0%)
Adenocarcinoma	11 (18.0%)	8 (22.2%)	3 (12.0%)
Surgical treatment	52 (85.2%)	30 (83.3%)	22 (88.0%)
Brachytherapy	52 (85.2%)	31 (86.1%)	21 (84.0%)
Radiotherapy (EBRT)	50 (82.0%)	32 (88.9%)	18 (72.0%)
Chemotherapy	27 (44.3%)	19 (52.8%)	8 (32.0%)

## Data Availability

Data are available upon request.

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
