# Peer review of "Tumor Mutational Burden in Cervical Cancer as Potential Marker for Immunotherapy Responders"

_cancers, 2025, doi:10.3390/cancers17182963_

Round 1
Reviewer 1 Report
Comments and Suggestions for Authors
The study investigated the role of Tumor Mutational Burden (TMB) as a potential biomarker for immunotherapy response in cervical cancer. By analyzing tumor samples from 61 cervical cancer patients (82% squamous cell carcinoma, 18% adenocarcinoma) through targeted next-generation sequencing, the researchers classified tumors as TMB-high (≥10 mutations/Mb) or TMB-low. High TMB was identified in 59% of patients but showed no significant correlation with disease stage or HPV16/18 status. However, TMB was higher in patients with lymph node involvement, diabetes, and HPV52 infection. The genetic analysis revealed frequent mutations in genes such as PIK3CA, ARID1A, KRAS, and PTEN. Microsatellite instability was rare (3.3%), and homologous recombination deficiency (HRD) occurred in over half of the cases but did not correlate significantly with TMB status. The study highlights the complexity and potential clinical relevance of TMB in cervical cancer, suggesting it may serve as a complementary biomarker to guide future immunotherapy strategies.
- Can you further justify the choice of ≥10 mutations/Mb as the cutoff for high TMB beyond FDA approval guidelines? Did you consider testing alternative thresholds specific to cervical cancer?
- Given the finding that high TMB did not correlate with advanced FIGO stages or HPV16/18 status, how would you propose clinicians practically utilize TMB as a predictive biomarker for cervical cancer treatment?
- How do you interpret the identification of genetic variants (e.g., PIK3CA:p.E453Q, ARID1A:p.Q487SfsTer132) typically associated with colorectal or endometrial cancers rather than cervical cancer? Could this suggest a shared pathway or cross-tumor significance?
- With tumor cell content ranging from 5% to 90%, how was potential variability in TMB calculation managed, and could lower tumor purity have significantly influenced your genomic analyses?
- Can you suggest possible biological explanations or previous literature supporting the observed inverse relationship between hypertension and lower TMB?
- Were variants of uncertain significance confirmed or validated using independent methods (e.g., Sanger sequencing)? If not, how might this affect the reliability of variant classification?
- Given the absence of correlation between HRD and TMB, how do you envision integrating these two independent biomarkers in clinical decision-making for personalized immunotherapy?
- Could you clarify how MSI was assessed in this study, and why might the observed rate (only 3.3%) be lower than expected based on other cancers?
- Were survival outcomes (overall survival or progression-free survival) different between patients classified as TMB-high versus TMB-low? If such analyses were not performed, could you explain this decision?
Author Response
Comment 1: Can you further justify the choice of ≥10 mutations/Mb as the cutoff for high TMB beyond FDA approval guidelines? Did you consider testing alternative thresholds specific to cervical cancer?
Response 1: Thank you for this remark, the cutoff of ≥10 mutations/Mb was selected based on the FDA-approved, pan-cancer definition of TMB-high validated in KEYNOTE-158 (Marabelle et al., Lancet Oncol 2020). To our knowledge, there are no cervical cancer-specific thresholds endorsed by regulatory agencies. We have clarified this in the revised Discussion and noted in the Limitations that future studies should refine disease-specific thresholds. In the same time, In NGS, Tumor Mutational Burden (TMB) is calculated based on the number of non-synonymous variants detected in the exonic regions of the panel. The calculation using is standardized as follows:
• High TMB is defined as >23 mutations per megabase (Mut/Mb)
• The panel covers 0.64 Mb of exonic regions, and the number of mutations is normalized accordingly.
This threshold (23 Mut/Mb) was internally validated for NGS, and different from the FDA's commonly used ≥10 Mut/Mb based on whole-exome sequencing. The difference arises from variations in panel size and target region content.
As a sensitivity analysis we now include the TMB of 23 Mut/Mb, please see methods' section "Bioinformatic Processing and Variant Annotation" changed and Figure 5 updated.
Comment 2: given the finding that high TMB did not correlate with advanced FIGO stages or HPV16/18 status, how would you propose clinicians practically utilize TMB as a predictive biomarker for cervical cancer treatment?
Response 2: We agree with the reviewer. The absence of correlation with FIGO or HPV16/18 suggests that TMB should not be used alone in early-stage disease. We now emphasize its role as a complementary biomarker for advanced disease and in combination with PD-L1, MSI, and TILs (please see p 10 lines 320-3)
Comment 3: How do you interpret the identification of genetic variants (e.g., PIK3CA:p.E453Q, ARID1A:p.Q487SfsTer132) typically associated with colorectal or endometrial cancers rather than cervical cancer? Could this suggest a shared pathway or cross-tumor significance?
Response 3: We agree this is noteworthy. These variants may reflect shared pathways (e.g., APOBEC-driven mutagenesis) across tumor types. We have added a paragraph to the Discussion highlighting this potential biological overlap. (please see p 10 lines 305-12)
Comment 4: With tumor cell content ranging from 5% to 90%, how was potential variability in TMB calculation managed, and could lower tumor purity have significantly influenced your genomic analyses?
Response 4: Thank you for this insightful remark; Since both TMB and MSI rely on SNV/INDEL detection, low tumor purity can lead to underestimation of both metrics. We acknowledge variability in tumor purity (5–90%) and have added this as a limitation. We recommend caution when interpreting samples with <20% tumor content, especially for INDEL-heavy assessments like MSI. While quality control filters were used, we did not perform purity-based normalization. (please see p. 3 lines 117-22 and p 10 lines 339-41).
Comment 5: Can you suggest possible biological explanations or previous literature supporting the observed inverse relationship between hypertension and lower TMB?
Response 5: We could not identify prior evidence linking hypertension to low TMB. We note in the Discussion that this observation may be incidental and requires validation. (please see p. 10 lines 320-23).
Comment 6: Were variants of uncertain significance confirmed or validated using independent methods (e.g., Sanger sequencing)? If not, how might this affect the reliability of variant classification?
Response 6: Thank you for this brilliant remark; The Variants of Uncertain Significance (VUS) were not confirmed using independent method. The decision not to validate the VUS through an independent method was influenced by several critical factors inherent to oncological diagnostics. In the context of cancer patients, timely therapeutic decisions are crucial. The necessity for rapid analysis to guide treatment options often limits the feasibility of extended validation procedures. Delays in confirming VUS variants could postpone potentially life-saving interventions.
The evaluation of Tumor Mutation Burden (TMB) as a biomarker requires comprehensive genomic profiling. Our approach utilizes a broad 344-gene panel, which inherently detects a large number of variants, including numerous VUS. The high volume of VUS identified makes individual validation of each variant impractical within the clinical timeframe.
While independent validation enhances confidence in variant classification, the overarching priority remains to provide actionable genomic data swiftly to inform patient management. Thus, the lack of validation for the specific VUS is a considered compromise aimed at balancing analytical rigor with the urgent needs of oncology care.
Comment 7: Given the absence of correlation between HRD and TMB, how do you envision integrating these two independent biomarkers in clinical decision-making for personalized immunotherapy?
Response 7: We agree that HRD and TMB are independent. We emphasize that only MSI is currently validated, and future research should explore gene-level mutation combinations rather than aggregate TMB; (please see p 10 lines 334-5)
Comment 8: Could you clarify how MSI was assessed in this study, and why might the observed rate (only 3.3%) be lower than expected based on other cancers?
Response 8: Thank you for the opportunity to clarify;
Microsatellite instability (MSI) in NGS is assessed using multiple conditions:
- Total number of SNV/INDEL mutations across all genes > 30
- INDEL Index > 20%
- INDEL Index = Proportion of INDEL mutations among total SNV/INDEL
Additional considerations include:
- Presence of AMC MSI marker (e.g., ACVR2A K437Rfs5*)
- Detection of pathogenic MMR gene mutations
- Evaluation of tumor purity
- Filtering of germline variants, false positives/negatives
- POLE mutations (pathogenic) are also taken into account for comprehensive interpretation.
Our observed MSI rate (3.3%) is consistent with TCGA data (<5%) and reflects HPV-driven rather than mismatch repair–driven tumorigenesis. This is now discussed. (please see p 10 lines 297-312).
Comment 9: Were survival outcomes (overall survival or progression-free survival) different between patients classified as TMB-high versus TMB-low? If such analyses were not performed, could you explain this decision?
Response 9: Thank you for this remark, although initially not intended because of small numbers, We have added an exploratory Kaplan–Meier analysis of OS and PFS by TMB status (Appendix Figure 1) and referenced it in the Results; (please see p 8 lines 228-30).
Reviewer 2 Report
Comments and Suggestions for Authors
This manuscript investigates the tumor mutational burden (TMB) and associated genomic features in a retrospective cohort of cervical cancer patients using panel-based next-generation sequencing. The authors present a comprehensive dataset that explores TMB in relation to clinical parameters, HPV subtypes, homologous recombination deficiency (HRD), and somatic mutation patterns. The study is timely and potentially impactful, given the growing interest in immunotherapy biomarkers for cervical cancer.
However, several aspects require significant revision. Notably, key associations (e.g., between TMB and nodal involvement, HPV52, diabetes) are derived from univariable analyses only and lack adjustment for potential confounders. The rationale for combining two different sequencing panels (ONCOaccuPanel and BRCAaccuTest) needs clarification, particularly since BRCA alterations are not central to cervical cancer pathogenesis. Additionally, although the manuscript references FDA-approved thresholds for TMB-high status, the authors should provide a more nuanced contextualization of the panel-based TMB values against gold-standard whole-exome sequencing and established immune biomarkers (e.g., PD-L1, TILs, MSI). Clearer legends and statistical annotations could strengthen several figures, and inconsistent terminology (e.g., “ca. planoepitheliale” vs “squamous cell carcinoma”) should be corrected throughout. Avoid repeating background data from the introduction in the discussion (e.g., vaccine coverage, global burden).
Line 60–66: Add a sentence summarizing why TMB may be especially complex in cervical cancer, e.g., HPV-induced mutagenesis vs. smoking, inflammatory microenvironment.
Line 71 (Patient cohort): Clarify how the 61 patients were selected—was this consecutive sampling?
Line 88–92 (Panels used):The dual use of BRCAaccuTest PLUS and ONCOaccuPanel is not justified. Why was BRCA panel used in cervical cancer?
Line 103–104 (TMB cutoff):FDA cutoff of ≥10 mutations/Mb is stated, but it would be good to mention the clinical basis for this threshold in cervical cancer (e.g., KEYNOTE-158 study).
Line 106–119 : Include whether multivariable analysis was performed. Only univariable is mentioned.
Clinical Characteristics (Lines 120–137)Reporting of "in-hospital deaths" is rare in sequencing studies. Consider removing or explaining relevance.
Genomic Landscape (Lines 138–187)Report SBS signatures numerically or compare with COSMIC Signature 2/13 (APOBEC-related), which is common in HPV-positive cancers.
Line 191:Note how MSI was measured (e.g., inferred from panel or PCR/IHC).
Line 236–254:Emphasize how the panel-based TMB correlates with WES-derived TMB in cervical tumors (if known).
Line 255–266:The discussion asserts that high TMB correlates with nodal involvement and diabetes, but without multivariable control, this claim is weak.
Line 267–273:Could explore therapeutic relevance—e.g., is HRD actionable in cervical cancer (e.g., PARP inhibitors)? Mention if BRCA mutations were found.
Use of “ca. planoepitheliale” is inconsistent with “squamous cell carcinoma.” Standardize terminology throughout.
Several typographical errors: e.g., “adenocardicona” (should be adenocarcinoma).
Improve clarity in describing data presentation (e.g., "12%" TMB in squamous is ambiguous—mean TMB? Proportion with high TMB?).
Author Response
However, several aspects require significant revision.
Comment 1: Notably, key associations (e.g., between TMB and nodal involvement, HPV52, diabetes) are derived from univariable analyses only and lack adjustment for potential confounders.
Response 1: Thank you for this great comment; we have indeed performed multivariable logistic regression adjusting for FIGO stage, histology, age, and HPV subtype that followed univariable associations; however sample size is limited and no association could have been confirmed in multivariable model; we acknowledge that in limitations section (please see p. 10 lines 320-23)
Comment 2: The rationale for combining two different sequencing panels (ONCOaccuPanel and BRCAaccuTest) needs clarification, particularly since BRCA alterations are not central to cervical cancer pathogenesis.
Response 2: Great comment, we explained that BRCAaccuTest was included to assess HRD-related genes, as part of exploratory analysis (p2-3 lines 91-95)
Comment 3: Additionally, although the manuscript references FDA-approved thresholds for TMB-high status, the authors should provide a more nuanced contextualization of the panel-based TMB values against gold-standard whole-exome sequencing and established immune biomarkers (e.g., PD-L1, TILs, MSI).
Response 3: We agree with the Reviewer. Indeed, the cutoff of ≥10 mutations/Mb was selected based on the FDA-approved, pan-cancer definition of TMB-high validated in KEYNOTE-158 (Marabelle et al., Lancet Oncol 2020). To our knowledge, there are no cervical cancer-specific thresholds endorsed by regulatory agencies. We have clarified this in the revised Discussion and noted in the Limitations that future studies should refine disease-specific thresholds.
In the same time, In NGS, Tumor Mutational Burden (TMB) is calculated based on the number of non-synonymous variants detected in the exonic regions of the panel. The calculation using is standardized as follows:
• High TMB is defined as >23 mutations per megabase (Mut/Mb)
• The panel covers 0.64 Mb of exonic regions, and the number of mutations is normalized accordingly.
This threshold (23 Mut/Mb) was internally validated for NGS, and different from the FDA's commonly used ≥10 Mut/Mb based on whole-exome sequencing. The difference arises from variations in panel size and target region content.
As a sensitivity analysis we now include the TMB of 23 Mut/Mb, please see methods' section "Bioinformatic Processing and Variant Annotation" changed and Figure 5 updated.
Comment 4: Clearer legends and statistical annotations could strengthen several figures, and inconsistent terminology (e.g., “ca. planoepitheliale” vs “squamous cell carcinoma”) should be corrected throughout.
Response 4: We totally agree with the Reviewer and corrected this throughout the manuscript.
Comment 5: Avoid repeating background data from the introduction in the discussion (e.g., vaccine coverage, global burden).
Response 5: Thank you for this remark, we have removed this part and updated references accordingly;
Comment 6: Line 60–66: Add a sentence summarizing why TMB may be especially complex in cervical cancer, e.g., HPV-induced mutagenesis vs. smoking, inflammatory microenvironment.
Response 6: Thank you, great idea, the respective sentence is now incorporated in the introduction section (p2 lines 62-64)
Comment 7: Line 71 (Patient cohort): Clarify how the 61 patients were selected—was this consecutive sampling?
Response 7: Indeed, the current cohort represents consecutive series of patients profiled between 2014–2021, please see (p 2 lines 74-7)
Comment 8: Line 88–92 (Panels used):The dual use of BRCAaccuTest PLUS and ONCOaccuPanel is not justified. Why was BRCA panel used in cervical cancer?
Response 8: Thank you again for the comment; BRCAaccuTest was included to assess HRD-related genes, as part of exploratory analysis (p2-3 lines 91-95)
Comment 9: Line 103–104 (TMB cutoff):FDA cutoff of ≥10 mutations/Mb is stated, but it would be good to mention the clinical basis for this threshold in cervical cancer (e.g., KEYNOTE-158 study).
Response 9: Thank you for this comment; we have updated the MS with the sensitivity threshold and explained the rationale behind choosing the previous one (please refer to response 3).
Comment 10: Line 106–119 : Include whether multivariable analysis was performed. Only univariable is mentioned.
Response 10: thank you for this comment; we now include this information, please refer to response 1 for details.
Comment 11: Clinical Characteristics (Lines 120–137)Reporting of "in-hospital deaths" is rare in sequencing studies. Consider removing or explaining relevance.
Response 11: We are grateful for this comment; as primarily gynecological surgeons we thought it was important to report on surgical complications and in-hospital outcomes; we leave the decision to the Editor.
Comment 12: Genomic Landscape (Lines 138–187) Report SBS signatures numerically or compare with COSMIC Signature 2/13 (APOBEC-related), which is common in HPV-positive cancers.
Response 12: Thank you for the comment; we now report the signatures numerically and also compare to COSMIC signatures (please see p. 5 lines 165-170)
Comment 13: Line 191: Note how MSI was measured (e.g., inferred from panel or PCR/IHC).
Response 13: we include details on MSI measurements in the methods section (p3. lines 117-21)
Comment 14: Line 236–254: Emphasize how the panel-based TMB correlates with WES-derived TMB in cervical tumors (if known).
Response 14: Indeed, a great comment; we include the following sentence into the discussion section: "
Although direct correlation data for our specific panel in cervical cancer is limited, studies have demonstrated that large hybrid capture panels like the one used here show strong correlation with WES-derived TMB across various cancer types, providing a reliable and practical alternative for clinical assessment [23, 25]. This alignment supports the validity of our TMB measurements"
Comment 15: Line 255–266:The discussion asserts that high TMB correlates with nodal involvement and diabetes, but without multivariable control, this claim is weak
Response 15: we agree and weaken the claim of association accordingly (please see p 9 line 281)
Comment 16: Line 267–273:Could explore therapeutic relevance—e.g., is HRD actionable in cervical cancer (e.g., PARP inhibitors)? Mention if BRCA mutations were found.
Response 16: We thank the reviewer for this insightful suggestion. We have now expanded the discussion on the clinical relevance of HRD in cervical cancer. As correctly implied, HRD is not currently an actionable biomarker in this disease, unlike in ovarian cancer. We explicitly state this and note that only a few pathogenic *BRCA1/2* mutations were found in our cohort. The text now clarifies that while a high rate of HRD was observed, its utility remains investigational and a subject for future clinical trials. Please see the revised paragraph in the Discussion section (p 9, lines 297-304).
Comment 17: Use of “ca. planoepitheliale” is inconsistent with “squamous cell carcinoma.” Standardize terminology throughout.
Response 17: Thank you for this remark, we have corrected this throughout the manuscript accordingly.
Comment 18: Several typographical errors: e.g., “adenocardicona” (should be adenocarcinoma).
Response 18: Thank you for spotting this error; we have now corrected these.
Comment 19: Improve clarity in describing data presentation (e.g., "12%" TMB in squamous is ambiguous—mean TMB? Proportion with high TMB?).
Response 19: We agree and have now corrected this part (please see p.7 215-6)